# Auxin-Induced *SaARF4* Downregulates *SaACO4* to Inhibit Lateral Root Formation in *Sedum alfredii* Hance

**DOI:** 10.3390/ijms22031297

**Published:** 2021-01-28

**Authors:** Dong Xu, Zhuchou Lu, Guirong Qiao, Wenmin Qiu, Longhua Wu, Xiaojiao Han, Renying Zhuo

**Affiliations:** 1State Key Laboratory of Tree Genetics and Breeding, Chinese Academy of Forestry, Beijing 100091, China; xudongzhuanyong@caf.ac.cn (D.X.); luzc@caf.ac.cn (Z.L.); qiaogr@caf.ac.cn (G.Q.); qiuwm05@caf.ac.cn (W.Q.); 2Forestry Faculty, Nanjing Forestry University, Nanjing 210037, China; 3Key Laboratory of Tree Breeding of Zhejiang Province, The Research Institute of Subtropical of Forestry, Chinese Academy of Forestry, Hangzhou 311400, China; 4National Engineering Laboratory of Soil Pollution Control and Remediation Technologies, Institute of Soil Science, Chinese Academy of Sciences, Nanjing 210008, China; lhwu@issas.ac.cn

**Keywords:** *SaARF4*, *SaACO4*, ethylene, auxin, *PINs*, lateral root

## Abstract

Lateral root (LR) formation promotes plant resistance, whereas high-level ethylene induced by abiotic stress will inhibit LR emergence. Considering that local auxin accumulation is a precondition for LR generation, auxin-induced genes inhibiting ethylene synthesis may thus be important for LR development. Here, we found that auxin response factor 4 (*SaARF4*) in *Sedum alfredii* Hance could be induced by auxin. The overexpression of *SaARF4* decreased the LR number and reduced the vessel diameters. Meanwhile, the auxin distribution mode was altered in the root tips and *PIN* expression was also decreased in the overexpressed lines compared with the wild-type (WT) plants. The overexpression of *SaARF4* could reduce ethylene synthesis, and thus, the repression of ethylene production decreased the LR number of WT and reduced *PIN* expression in the roots. Furthermore, the quantitative real-time PCR, chromatin immunoprecipitation sequencing, yeast one-hybrid, and dual-luciferase assay results showed that *SaARF4* could bind the promoter of 1-aminocyclopropane-1-carboxylate oxidase 4 (*SaACO4*), associated with ethylene biosynthesis, and could downregulate its expression. Therefore, we concluded that *SaARF4* induced by auxin can inhibit ethylene biosynthesis by repressing *SaACO4* expression, and this process may affect auxin transport to delay LR development.

## 1. Introduction

An optimal root system is essential for plant health and productivity [1]. Under abiotic stress in particular, the survival of plants is determined by the plasticity of root growth [2]. Promoting the formation of lateral roots (LRs) and adventitious roots (ARs) is an adaptive strategy used by plants in response to environmental changes [3,4]. These activities are referred to as the stress-induced morphogenic response (SIMR) [5]. LRs are an important component of SIMR [6]. It is reported that more LRs are induced under phosphate-deficiency or a salty environment [7,8]. Moreover, LR formation appears to be dependent on the strength of the abiotic stress. The generation of LRs will be promoted under low salt stress, whereas the LR number will be seriously decreased under high salt stress [3]. Significant differences in the structure of the root system can also be found when culturing plants using low and high nutrient solutions [9]. These results indicate that LR formation should be the result of an interaction between the in vivo and in vitro signals of plants. Furthermore, local auxin accumulation (LAA) is a prerequisite for LR formation [10]. However, the detailed mechanisms of this prerequisite remain to be further elucidated. In particular, which genes function in the process of LAA formation, and how do these genes incorporate environmental and endogenous signals to control LR development?

The interaction between auxin and ethylene is the basis for root development [11]. Ethylene can adjust auxin transport and can affect the LAA process [12,13]. Therefore, low levels of ethylene can be helpful for LR initiation [11]. However, abiotic stress can induce large amounts of ethylene, while high levels of ethylene will repress LR formation [11]. In other words, genes restricting ethylene synthesis may contribute to LR development and LAA is a prerequisite for LR development. Thus, the auxin-induced genes limiting ethylene biosynthesis may play important roles in LR formation under abiotic stress. These types of genes should have an ability to connect the auxin pathway and ethylene pathway. According to previous studies, auxin response factor (*ARF*) genes and 1-aminocyclopropane-1-carboxylate oxidase (*ACO*) genes (transforming ethylene precursor to ethylene) [14] may be a bridge between auxin and ethylene. For example, *MdARF5* from *Malus domestica* can promote ethylene biosynthesis to initiate apple fruit ripening by regulating *MdACO1* [15]. Similarly, *SlARF7* from *Solanum lycopersicum* can trigger *SlACO4* expression to influence the set and early development of tomato fruits [16]. These studies mainly focused on how *ARFs* adjust ethylene production to control fruit development by regulating *ACO* expression. Although ethylene is also important for LR development, whether *ARFs* can dominate ethylene synthesis to modify LR development by regulating *ACO* expression has not been thoroughly elucidated.

The *miR390*/*TAS3*/*ARFs* module is well-known for its functions in controlling LR development [17,18,19]. Although this module is highly conserved among different species, slight differences can also be found. For instance, the changes in LR density in *Arabidopsis* did not reach a significant level when overexpressing *miR390* [18] whereas LR number was highly increased in poplar [19]. However, it is undeniable that this module plays important roles in LR development. *ARF2*, *ARF3*, and *ARF4* are involved in this module, and these three genes are all transcription repressors [20]. Among them, the overexpression of *ARF4* inhibits LR development under salt stress, indicating that *ARF4* can function in root development under abiotic stress [19]. Moreover, in our previous studies, *SaARF4* was found to be an important hub gene in a co-expression network under cadmium (Cd) stress [21]. Thus, *SaARF4* was selected as the candidate gene for elucidating the mechanisms behind its functions in LR development.

In this study, the Cd/zinc co-hyperaccumulator *Sedum alfredii* Hance (HE) [22] was selected as the plant material to confirm (1) whether *SaARF4* can regulate LR development in HE; (2) whether there is an *ACO* gene that is a downstream gene for *SaARF4*; and (3) the detailed mechanisms by which *SaARF4* regulates *ACO* to change LR development.

## 2. Results

### 2.1. Overexpression of SaARF4 Inhibited the Development of the Vessels, LRs, and ARs

In the *miR390*/*TAS3*/*ARFs* module, microRNA390 (*miR390*) can effectively decrease *ARF4* (the target of *miR390*) expression, based on previous studies [17,18,19]. Therefore, *MIR390* (Appendix A) and SaARF4 were cloned from HE DNA and cDNA sequences, respectively. They were then separately introduced into HE to generate the overexpression transgenic lines miR390-OE and SaARF4-OE. We verified the cleavage site of *miR390* in *SaARF4* (Appendix A), and the overexpression of *miR390* reduced the *SaARF4* expression (Appendix A). However, we did not observe any significant differences in the total length or number of roots in miR390-OE compared with the wild-type (WT) plants (Appendix A). In sharp contrast, overexpressing *SaARF4* seriously delayed the development of ARs and LRs (Figure 1A). In detail, the generation of ARs in the WT was earlier than that in SaARF4-OE (Appendix A). Moreover, the number of ARs in SaARF4-OE was lower than that of WTs after the plants were cultured for about 21 d (Figure 1B, *p* < 0.05). The number of LRs in SaARF4-OE was only 35.4% of that in the WT (Figure 1C, *p* < 0.01). Similarly, the lengths of the ARs and LRs in SaARF4-OE were smaller than those in the WT (Figure 1D,E, *p* < 0.01). Cross sections of the SaARF4-OE stems were made and stained with HCl-phloroglucinol to identify whether any changes could be observed in the structures of the SaARF4-OE stems. The results indicated that aberrant vessels were formed in SaARF4-OE stems (Figure 1F). The diameter, number, and total area of the SaARF4-OE vessels were all decreased compared with those of the WT (Figure 1G–I). For example, the diameters of the SaARF4-OE vessels were only 60.0% of those in the WT (Figure 1G).

### 2.2. SaARF4 Is Expressed in Vascular Tissue and Is Upregulated by Auxin

LRs are generally generated at the LAA sites. The main characteristic of these sites compared with other locations in the rots is a higher auxin concentration. Therefore, analyzing how auxin affected the *SaARF4* expression is important for clarifying its roles at these sites. Meanwhile, as the expression sites of a gene are closely related to its functions, determining the expression sites of *SaARF4* was beneficial for the following functional analyses. Therefore, we fused the *SaARF4* promoter to a β-glucuronidase (GUS) reporter gene, and the recombinant genes were introduced into HE to produce *ProARF4*::*GUS* transgenic lines. *GUS* expression was observed in the roots, stems, and leaves of transgenic plants after histochemical staining (Figure 2A). Meanwhile, *GUS* signals were also observed in the vascular tissue of the stems (Figure 2B). Specifically, strong preferential expression was present in the surrounding areas of the vessels in the primary xylem (Figure 2C and Appendix A). However, *GUS* was weakly expressed in the phloem (Figure 2B, the red arrow). A similar pattern was also found in the leaves (Figure 2D,E). This expression site of *SaARF4* was consistent with its function of changing vessel diameter (Figure 1F). Moreover, as changes in vascular tissues may alter polar auxin transport [23,24], the effect of *SaARF4* on changing the vessel diameter indicated that its functions may affect auxin transport.

Naphthylphthalamic acid (polar auxin transport inhibitor, NPA) [25] and indoleacetic acid (IAA) were used to confirm the relationship between *SaARF4* and auxin. We found that, in the control plants (without any treatment), *GUS* was expressed in the tips and other places of the roots (Figure 2G, the red arrow). However, when inhibiting polar auxin transport using NPA, the activity of the *SaARF4* promoter was only restricted to the root meristem zone that can produce auxin by itself (Figure 2H, the red arrow). By contrast, the application of IAA enhanced the GUS signals (Figure 2I, the red arrow). Consistent with this result, the use of IAA significantly increased the *GUS* expression in *ProARF4*::*GUS*. (Figure 2F, *p* < 0.05). These results indicated that *SaARF4* can be induced by auxin. In other words, *SaARF4* expression at the LAA sites with a higher auxin concentration should be higher than those in the other parts of the roots.

### 2.3. SaARF4 Decreased Ethylene Content and Adjusted PIN Expression

*ACO* genes are the key enzymes that convert ACC to ethylene. Thus, before identifying the relationship between *SaARF4* and *ACOs*, we wondered whether *SaARF4* could affect ethylene production. Moreover, according to the above analysis, we also needed to further evaluate whether *SaARF4* could regulate auxin transport as well as the relationships among *SaARF4*, ethylene, and auxin transport. Thus, we measured the ethylene production of WT and SaARF4-OE under Cd stress. The results showed that ethylene was increased at 0–2 h in both WT and SaARF4-OE (Figure 3A). However, the ethylene content of SaARF4-OE did not increase significantly compared with that of WT after 2 h. To identify if overexpressing *SaARF4* could change the mode of auxin distribution, *ProDR5*::*GUS* was introduced into WT and SaARF4-OE using transgenic technology. We found that the distribution of GUS signals in the DR5pro:GUS-SaARF4-OE root tips was significantly different from that of DR5pro:GUS-WT (Figure 3B). Auxin in the root tips is partially transported by PIN proteins [26], especially *PIN1*, *PIN3*, and *PIN7*. In order to further identify whether *SaARF4* could change auxin transport, we measured the *PIN* expression in SaARF4-OE and WT. A total of 16 *PINs* were detected in the HE genome. Compared with those in the WT, eight *PINs* in SaARF4-OE were repressed (Figure 4C, *p* < 0.05). Among them, the expressions of *SaPIN1*, *SaPIN2*.*3*, *SaPIN3*.*2*, *SaPIN4*, *SaPIN5*, and *SaPIN7*.*2* were highly decreased (*p* < 0.01).

*SaARF4* can reduce the production of ethylene (Figure 3A), and ethylene was previously reported to be able to regulate *PIN* expression [12,27]. Therefore, we next determined whether these *PINs* were directly influenced by *SaARF4* or indirectly regulated by *SaARF4* via ethylene. Pyrazinamide (PZA, ethylene biosynthesis inhibitor) can effectively inhibit the activities of *ACO* enzymes to block ethylene production [28]. Therefore, WT was treated with PZA to determine whether the *PIN* expression was affected by ethylene. The results indicated that the expression patterns of *PINs* in WT with PZA treatment were highly similar to those of SaARF4-OE (Figure 3C,D), especially the *PINs* that were reported to be related to auxin transport in the roots [26] (the red boxes). Meanwhile, the promoter elements of these *PINs* were analyzed using the PlantCARE website (http://bioinformatics.psb.ugent.be/webtools/plantcare/html/), and the results indicated that the AuxRE element (TGTCTC, the special binding site for *ARFs*) was not found in these *PIN* promoters (Appendix A). Therefore, we speculated that *SaARF4* regulated the *PINs* by influencing ethylene production. In other words, the effect of *SaARF4* on the root development may be mediated by restricting ethylene production. In order to further confirm this conclusion, we need to identify whether inhibiting ethylene (using PZA) could influence root development in the WT under Cd stress. The results indicated that the root development of the WT under Cd stress was highly inhibited by PZA treatment (Figure 3E). In detail, the LRs were 19.0% of that in the control plants (without PZA treatment) and the ARs also decreased by 49.0% under PZA treatment (Figure 3F). In summary, in combination with the functions of ethylene in *PIN* expressions, we drew a basic conclusion that ethylene induced by Cd stress plays important roles in root development by regulating *PIN* expression while *SaARF4* acted on root development by regulating ethylene production.

### 2.4. SaARF4 Negatively Regulated Its Downstream Gene, SaACO4

It has been confirmed in previous studies that *ARFs* can control *ACOs* to influence ethylene [15,16]. Our results indicated that root development inhibition was detected in SaARF4-OE and that similar phenotypes were observed in the WT with PZA treatment (Figure 1 and Figure 3). In order to identify which genes *SaARF4* regulates in ethylene biosynthesis, we performed ChiP (chromatin immunoprecipitation)-seq of *SaARF4*. The results of the ChiP-seq (Appendix A) indicated that the AuxRE element in *SaACO4* was a possible binding site of *SaARF4* (the predicted binding site is indicated in Figure 4B). The dual-luciferase assay system and yeast one-hybrid (Y1H) were used to identify the relationship between *SaARF4* and *SaACO4*. The construction of relevant vectors is indicated in Figure 4B. The results showed that *SaARF4* could effectively rescue the auxotrophic phenotype of yeast (Figure 4C), indicating that *SaARF4* can bind to the AuxRE element of the *SaACO4* promoter. In the dual-luciferase assay system, the fluorescence intensity was seriously decreased after adding SaARF4 (Figure 4D). On the contrary, when removing AuxRE from the *SaACO4* promoter (Figure 4B), the fluorescence intensity of the dual-luciferase assay system was increased (Figure 4D,F), and these results were also verified by qRT-PCR (Figure 4D). In addition, a 20-fold lower expression level of *SaACO4* was detected in SaARF4-OE than in WT (Figure 4A). In summary, *SaARF4* can negatively regulate *SaACO4* by binding to the AuxRE in the *SaACO4* promoter, and this conclusion was consistent with the expression level of *SaACO4* in SaARF4-OE (Figure 4A).

## 3. Discussion

To date, *ARF2*, *ARF3*, *ARF4*, *ARF5*, *ARF6*, *ARF7*, *ARF8*, and *ARF19* have been reported to be associated with LR development [18,29]. Their downstream genes identified in recent years mainly function in the early developmental stages of LRs [30,31,32], while the *SHY2*/*IAA3–ARFs* module is related to auxin signals [33]. These studies indicated that *ARFs* may profoundly affect almost every aspect of LR development. LAA is a basis for respecifying root pericycle cells into LR founder cells [34], and auxin transport is the precondition for LAA [35]. Understanding whether *ARFs* are involved in auxin transport and how their functions are achieved is an important aspect for understanding the LR developmental mechanism. In this study, we found that *SaARF4* may alter auxin transport to delay LR development. First, the vessel diameter in SaARF4-OE was significantly smaller than that in WT (Figure 1G), and this phenotype was similar to those induced by a polar auxin transport inhibitor [36]. Second, differences in auxin distribution were found in the root tips by introducing *ProDR5*:*GUS* into WT and SaARF4-OE (Figure 3B). Third, *PINs* related to auxin transport in the roots [26] were seriously downregulated (Figure 3C). The decline in ethylene content caused by *SaARF4* regulating *SaACO4* may be an important reason for the variations in auxin transport. Two lines of evidence supported our judgement: the expression patterns of *PINs* in the roots of WT were similar to those of SaARF4-OE at the application of PZA (Figure 3C,D), and moreover, PZA could effectively inhibit the generation of LRs of the WT (Figure 3). In previous studies, *MdARF5* in apples effectively upregulated *MdACO1* to enhance ethylene production [15], and *SlARF7* in tomato positively regulated *SlACO4* to influence fruit development [16]. These studies mainly focused on the roles of ethylene in fruit initiation [16] or ripening [15]. Together, our study and previous studies have demonstrated the roles of *ARFs* and *ACOs* in connecting the pathways of auxin and ethylene.

It is a common phenomenon that multiple hormones interact with each other to regulate the developmental process of plants [37,38]. An important function of the interactions is to transform in vivo/in vitro signals into developmental signals [39]. Furthermore, these regulatory processes mainly alter the transport or content of auxin to modify plant morphologies [13]. For example, ABA upregulated *PIN2* under osmotic stress, resulting in decreased auxin content in the root meristem leading to root growth inhibition [40]. The interactions between auxin and ethylene act in many developmental aspects of plants, such as apical hook development [41], fruit ripening [42], and primary root elongation [43]. It is worth noting that auxin and ethylene regulate LR development based on a certain threshold level [11]. For example, a low level of ACC can promote LR initiation while higher doses of ACC will inhibit LR development [11,44]. The dose response reflects the complexity and flexibility of developmental regulation. Therefore, keeping a low ethylene content level is important for LR development, especially under abiotic stress. In this study, *SaARF4* induced by auxin significantly downregulated *SaACO4* expression (Figure 4A) and thus decreased ethylene production (Figure 2 and Figure 3). Based on these results, the ethylene content at the sites of LAA should be lower than that at other sites of the roots. As ethylene can accelerate the IAA transport rate [45,46], the decline in ethylene could be helpful for maintaining local auxin concentration. This may explain why LAA is a precondition for LR development. Ethylene increases IAA transport by enhancing the expressions of *PIN3* and *PIN7* to inhibit LR development in Arabidopsis [12]. In line with these studies, we also found that *SaPIN1*, *SaPIN3*.*2*, and *SaPIN7*.*2* were significantly decreased in the SaARF4-OE lines compared with those in WT (Figure 3C). Moreover, the expression patterns of these genes in SaARF4-OE were highly similar to those of WT treated with PZA (Figure 3C,D, the red wireframes). These results supported that *SaARF4* can downregulate these *PINs* by inhibiting ethylene production. In summary, more ethylene induced by abiotic stress will enhance *PIN* expression, and this process may not be conducive to LAA. *SaARF4* can effectively decrease ethylene production. Therefore, the functions of *SaARF4* will play important roles in LR development under abiotic stress.

In conclusion, we mainly described the roles of *SaARF4* and *SaACO4* in the sites of LAA of roots under abiotic stress in this study. A model involving *SaARF4* and *SaACO4* was built to clearly illustrate the mechanisms behind LR development. Although ethylene was induced at the beginning of Cd stress, the ethylene content was still at a low level. At this time, *PIN* expression upregulated by ethylene can be helpful for LAA (Figure 5A). With auxin accumulation, a large amount of auxin enhanced *SaARF4* expression (Figure 5B), following which *SaACO4* was seriously inhibited by *SaARF4*. Thus, ethylene production will be downregulated at the sites of LAA. The declined ethylene can repress *PIN* expression, and this process can effectively prevent the outflow of auxin from LAA sites (Figure 5C). Therefore, LRs tend to be generated at the sites of LAA. For this reason, more LRs grew in *S*. *alfredii* at a low level of Cd stress [22]. However, if the content or intensity of ethylene induced by abiotic stress exceeds the range that *SaARF4* could adjust to, the generation of LRs may also be inhibited. This may explain why LRs are inhibited under high levels of abiotic stress [22].

## 4. Materials and Methods

### 4.1. Plant Materials and Growth Conditions

The *S*. *alfredii* (HE) plants were cultured in our laboratory. The growth conditions and hydroponic experiments were as described previously [47].

### 4.2. Plasmid Construction

The open reading frame and 2 kb promoter of *SaARF4* were amplified from the cDNA and genomic DNA of HE, respectively. These products were cloned into pDONR222 and then recombined into pK2GW7.0 and pMDC164 to produce *35S*::*SaARF4* and *ProARF4*::*GUS*, respectively. Moreover, the *miR390* precursor (*MIR390*) was cloned from the DNA, and the precursor was also recombined into the Pk2GW7.0 vector to produce *35S*::*miR390*. Meanwhile, the complete coding sequence of *SaARF4* was assembled into pGADT7-Rec (for theY1H) and pGreenII 62-SK (for the dual-LUC reporter system) using the sites of SmaI/BamHI and EcoRI/HindIII, respectively. The promoter sequence (approximately 500 bp, *ProSaACO4*) of *SaACO4* was amplified from DNA. After that, overlapping PCR was used to remove the AuxRE element [15] from *ProSaACO4* to produce *ProSaACO4m*, which was then inserted into the vector of pGreenII 0800-LUC using SalI/BamHI to create *ProSaACO4*::*LUC*. The *ProDR5*::*GUS* vector was consistent with that in previous reports [48]. The primers used are listed in Appendix A.

### 4.3. Plant Transformation

The vectors containing *35S*::*SaARF4*, *35S*::*miR390*, *ProARF4*::*GUS*, and *ProDR5*::*GUS* were introduced into EHA105 *Agrobacterium tumefaciens* strain by electroporation. Subsequently, the *A. tumefaciens* containing the *35S*::*SaARF4*, *35S*::*miR390*, and *ProARF4*::*GUS* vectors was used to infect the HE calluses. After obtaining the overexpression transgenic lines of *SaARF4* (SaARF4-OE), the *A. tumefaciens* containing the *ProDR5*::*GUS* vector was used to infect the WT and SaARF4-OE lines. The *A. tumefaciens*-infected method was performed as described before [49] with minor modifications. The differentiation medium was a Murashige and Skoog medium (MS) + 2 mg·L^−1^ 6-benzylaminopurine + 0.3 mg·L^−1^ 1-naphthaleneacetic acid, and the rooting medium was 1/2 MS + 2 mg·L^−1^ 3-indolebutyric acid. The use of antibiotics is varied with the different vectors. The concentrations of kanamycin and hygromycin were 30 mg·L^−1^ and 20 mg·L^−1^, respectively. WT and transgenic plants were subsequently transplanted into soil for further detection and experimentation.

### 4.4. qRT-PCR

Total RNAs were extracted from the roots of WT and transgenic lines (with three different lines) using an RNA extraction kit (RNAprep Pure Plant Kit, TIANGEN, Dalian, China). Next, first-strand cDNAs were generated using a cDNA synthesis kit (PrimeScript™ RT Master Mix, TAKARA, Beijing, China). These cDNAs were utilized for qRT-PCR using TB Green reagent (TB Green™ Premix Ex Taq™, TAKARA, Dalian, China). Sequence Processing and Data Extraction (SPDE) was used for the batch design of the qRT-PCR primers [50]. All of the primers, including that of the reference gene (*UBC9*), are listed in Appendix A.

### 4.5. Identification of the Members of PIN Gene Family

The *PIN* gene family members were extracted from genomic files using the related pfam file (PF03547.18) on the SPDE software platform. These genes were retested by NCBI-Blast (https://blast.ncbi.nlm.nih.gov/Blast.cgi) and named after their homologous genes in *Arabidopsis* using the TAIR website (https://www.arabidopsis.org/). The genes with low expression levels (ct value > 30 in qRT-PCR experiment) in the roots were then also removed.

### 4.6. Histochemical Analyses and Tissue Section

The above 2nd–5th leaves and 3rd internode of the stems of *ProARF4*::*GUS* were collected for tissue section. β-glucuronidase (GUS) staining was performed as described previously [51], and three independent transgenic lines were used for GUS staining. The cross sections of WT and SaARF4-OE were made as described before [52], and the sections were stained by HCl-phloroglucinol (three biological replicates).

### 4.7. Auxin/Ethylene Inhibitors Treatment and IAA Application

The selection and application of NPA [26] and PZA [28] were based on published studies. In detail, six transgenic lines of *ProARF4*::*GUS* cultured under hydroponic conditions for two weeks were treated with 10 μM NPA and 20 μM IAA for 7 d. The WT plants were cultured hydroponically in the presence and absence of 100 μM PZA for about 14 d with three biological replicates.

### 4.8. Measurement of Ethylene Content

Cadmium treatment (as stated above) was conducted using equal qualities of SaARF4-OE and WT (3 g, at least three biological replicates) for 0–5 h. The ethylene content was measured using an F-900 Portable Ethylene Analyzer (Felix instruments, Camas, WA, USA) as in Lerud et al. (2019) [53].

### 4.9. Y1H Assay

The coding region of *SaARF4* was inserted into the vector of pGADT7-Rec (Clontech) to produce pGADT7-rec-SaARF4. A 20 bp fragment (as a unit) with the AuxRE motif was cloned from the *SaACO4* promoter. Four tandem copies of the unit were constructed and cloned into pHIS2 vectors to produce pHIS2-ACO4 [54]. The pGADT7-rec-SaARF4/pHIS2-ACO4 was co-transformed into AH109 (yeast strain) using the LiAc-PEG3350 method [55]. SD-Leu-Trp plates were used for transformant selection, and SD-Leu-Trp-His plates supplemented with 30 μΜ 3-AT (3-amino-1,2,4-triazole) were utilized for testing the interactions. Three biological replicates and three technical repeats were used for this process. The primers are listed in Appendix A.

### 4.10. ChIP Assays

The expression and purification of the *SaARF4* protein were performed as described before [56] with minor changes. After cloning the coding sequence of *SaARF4*, the sequence was inserted into the NotI and SbfI sites of the pMAL-c5x vector to generate pMAL-c5x-SaARF4. Then, the recombinant vector was transformed into *Escherichia coli* Arctic Express, and the recombinant protein was expressed under the condition of 0.5 mM β-D-thiogalactopyranoside (IPTG) at 20 °C. After that, the purified protein was used as the antigen to produce the mouse monoclonal antibody (AbMART). Western blot was performed for detecting the specificity of the antibody (Appendix A) [57]. The basic process of the ChIP assay refers to the method stated by Landt et al. (2012) [58]. In detail, approximately 4 g of SaARF4-OE leaves were collected for ChIP-seq. The chopped leaves were placed into 1% formaldehyde and vacuum-infiltrated for 15–30 min. These materials were ground in liquid nitrogen. After that, the chromatin was broken into 100–500 bp DNA fragments by sonication. The ChIP experiment was performed using SaARF4-antibody, and protein-DNA-antibody complexes were pulled down and digested using 2 μL proteinase K to obtain the product. Then, the product was purified to obtain the immunoprecipitated DNA. After testing the concentration and purity of DNA using Q-bit, the immunoprecipitated DNA (6 ng) was used for sequencing. The sequencing results were aligned into the genomic data of HE. The final ChIP-seq result is indicated in Appendix A.

### 4.11. Transient Transcriptional Activities Assay

The transient transcriptional activity assays were performed as described previously [59]. The corresponding vectors were constructed as stated above. A Lumazone imaging system (Mag Biosystems, Tucson, AZ, US) was used for the image acquisition of luciferase. Image processing was accomplished using the software ImageJ (https://imagej.net/Welcome). The qRT-PCR was conducted for testing the expression level of Luciferase and Rluc, and the Rluc that was driven by 35S promoter was chosen as the reference gene.

### 4.12. Identification of the Cleavage Sites of miR390 on SaARF4 Using RLM-RACE

The RLM-RACE was performed using the protocol of 5’RLM-RACE in the FirstChoice^®^ RLM-RACE Kit (https://www.thermofisher.com/order/catalog/product/AM1700M#/AM1700M).

### 4.13. Statistical Analysis

At least three biological replicates and three technical repeats were performed for each assay. SPSS 20.0 (IBM Corp., Armonk, NY, USA) was used for statistical analysis (https://www.ibm.com/cn-zh). The method of one-way ANOVA was used for significance analysis. All data are showed as means ± SD. The least significant difference (LSD) test was applied to analyze the differences at the 0.05 or 0.01 probability levels (* *p* < 0.05 and ** *p* < 0.01). The histograms were generated by Excel (Microsoft Corp., Albuquerque, NM, USA), and images were processed by Abode Illustrator and Abode Photoshop (Abode Systems).

## 5. Conclusions

In this study, we demonstrated a possible function of *SaARF4* in LR development. *SaARF4* was induced by auxin at the sites of LAA. The expression of *SaARF4* was enhanced, and *SaACO4* was seriously inhibited. The ethylene content was thus decreased, which could be helpful in the maintenance of local auxin concentration. This regulatory process may play important roles in LR development, especially under abiotic stress.

## Figures and Tables

**Figure 1 ijms-22-01297-f001:**
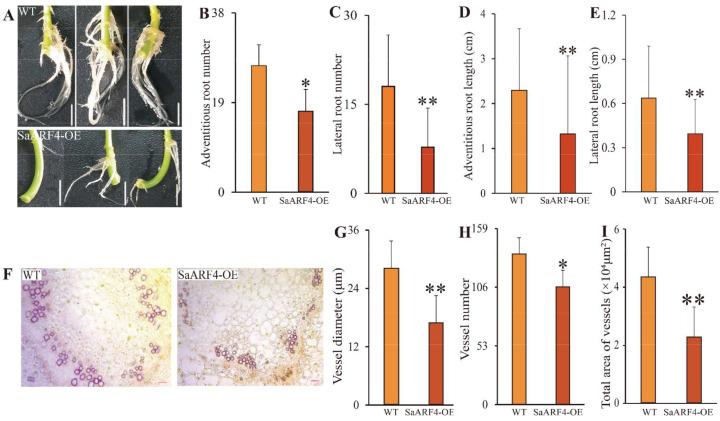
Changes in vessels, and lateral and adventitious roots of the SaARF4-OE lines: (**A**–**E**) the influence of *SaARF4* on the number and length of the lateral and adventitious roots (three biological replicates of the wild-type (WT) and SaARF4-OE each); (**F**) cross sections of the stems stained by HCl-phloroglucinol to identify the differences between the WT and SaARF4-OE lines (three biological replicates each); and (**G**–**I**) the changes in total area, number, and diameter of the vessels between the WT and SaARF4-OE lines. * *p* < 0.05; ** *p* < 0.01.

**Figure 2 ijms-22-01297-f002:**
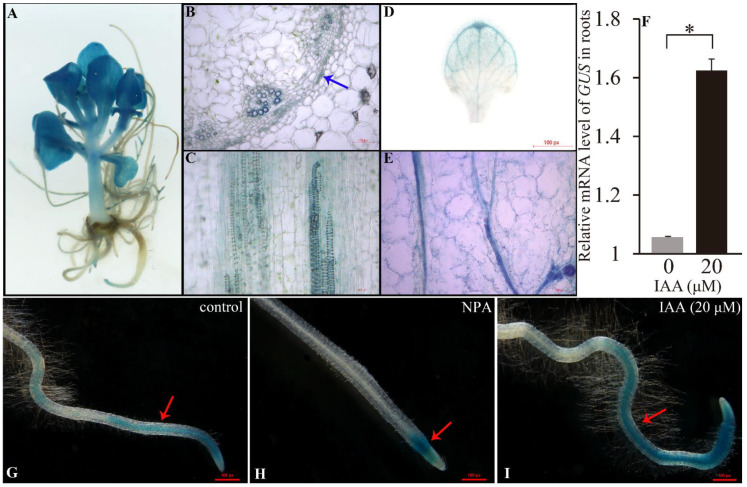
*SaARF4* was expressed in the vascular tissues and induced by auxin. (**A**) The β-glucuronidase (GUS) signals can be detected in the roots, stems, and leaves of *ProARF4*::*GUS*. The cross sections (**B**) and longitudinal sections (**C**) of stems were made for identifying the expression location of SaARF4. (**D**) The expression location of *SaARF4* in the leaves. (**E**) The expression location of *SaARF4* in longitudinal sections of the leaves. (**G**) The distribution of GUS signal in *ProARF4*::*GUS* roots without any treatment. (**H**,**I**) Naphthylphthalamic acid (NPA) (10 μM) and indoleacetic acid (IAA) were used to determine whether the changes in auxin content affected *SaARF4* expression in the roots. (**F**) The expression level of *ProARF4*::*GUS* under IAA treatment was detected by qRT-PCR based on three biological replicates and three technical replicates. The red arrows indicate GUS signals. * *p* < 0.05.

**Figure 3 ijms-22-01297-f003:**
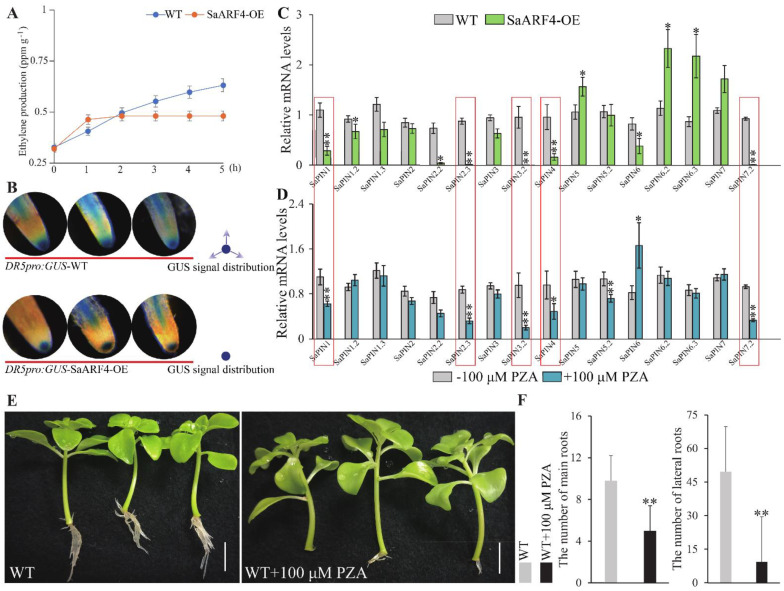
*SaARF4* affected ethylene production and auxin transport. The ethylene production of WT and SaARF4-OE, during 0–5 h under Cd stress is shown in (**A**). *ProDR5*:*GUS* was transformed into the WT and SaARF4-OE lines to identity the auxin distribution model (**B**). Most of the *PINs* related to auxin transport in the roots were influenced by the overexpression of SaARF4 (**C**). Moreover, similar expression modes could be obtained by applying pyrazinamide (PZA) to the WT (**D**). Under Cd stress (50 μM CdCl_2_), the number of adventitious roots (Ars) and lateral roots (LRs) under the influence of PZA were lower than those of the control plants (**E**,**F**). These assays were conducted with 100 μM PZA. * *p* < 0.05; ** *p* < 0.01.

**Figure 4 ijms-22-01297-f004:**
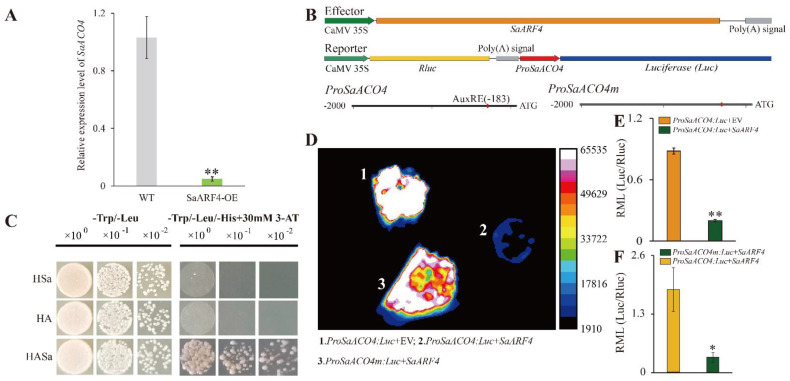
*SaARF4* directly downregulated *SaACO4*. Quantitative RT-PCR was conducted with three biological replicates and technical replicates to identify the effect of *SaARF4* overexpression on the expression of *SaACO*4 (**A**). Vectors in the dual luciferase assay system were performed (**B**). The results of the Y1H indicated physical binding between *SaARF4* and the *SaACO4* promoter (**C**). HSa, pHIS2 + pGADT7-rec-SaARF4; HA, pHIS-AuxRE + pGADT7-rec; HASa, pHIS2-AuxRE + pGADT7-rec-SaARF4. The intensity of the fluorescence indicates the promoter activities of *SaACO4* (**D**) in three biological replicates. Quantitative RT-PCR with Ren as the reference gene was conducted to further verify the results of the dual luciferase assay system (**E**,**F**). RML, relative mRNA level. * *p* < 0.05; ** *p* < 0.01.

**Figure 5 ijms-22-01297-f005:**
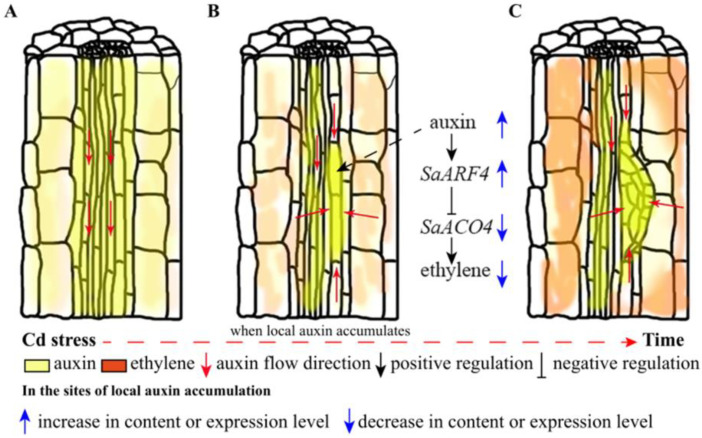
Model of *SaARF4* including *SaACO4* and ethylene under Cd stress in the LR developmental process: under normal conditions, auxin was evenly distributed in the roots (**A**); at the beginning of Cd stress, the production of ethylene was enhanced and promoted the process of local auxin accumulation (LAA). Then, the expression of *SaARF4* was enhanced with auxin accumulation, which gradually repressed *SaACO4* expression and finally inhibited ethylene production (**B**); the decline in ethylene production may be helpful for maintaining the auxin concentration at the sites of LAA, and this process may provide the auxin conditions for LR development (**C**).

## Data Availability

The data that support the findings of this study are available from the corresponding author upon reasonable request.

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
