# Peer review of "Auxin-Induced SaARF4 Downregulates SaACO4 to Inhibit Lateral Root Formation in Sedum alfredii Hance"

_ijms, 2021, doi:10.3390/ijms22031297_

Round 1
Reviewer 1 Report
Xu et al connect SaARF4 to ethylene production in the control of lateral and adventitious roots in the plant Sedum alfredii.
Whereas the study is of interest to the auxin community, there are large structural issues with this manuscript. In general, the methods are lacking specifics and clarity. It is not clear how many plants were used for any of the root studies or GUS studies presented throughout the papers. The number of independent transgenic lines that share the phenotypes should be stated and tested if possible. The number of individuals assayed for root growth or GUS staining experiments should be stated clearly so that the data may be assessed. In addition p-values are presented throughout the manuscript with no explanation for how they were determined. Are these the results of t-tests?
Further, Cd stress is used for the data presented in Figure 4 and in Figure 1A but is not mentioned in the root assays in Figure 1. Is Cd stress required for all phenotypes presented in this manuscript or just those presented in Figure 4. If Cd is only used for those experiments presented in Figure 4, then how can I compare the results between figures 1 and 4. For example the number of lateral roots presented in the wild-type condition in figure 1 is far lower than those presented in Figure 4 and more closely matches the number of lateral roots under PZA treatment. If these are under the same growth conditions, then this suggests to me that the later root growth phenotypes are highly variable.
There is also missing data from this manuscript. Figure 5 is mentioned in the text but it totally absent from my version. In addition, there are methods for a ChIP experiment with no explanation of how it conducted or presentation of the results that show SaARF4 binding to the SaACO4 promoter region. Is SaARF4 binding found at the other ACO promoters? I cannot say because the data is not presented or otherwise made available.
There are issues with the ChIP experiment as well. There is no validation that the anti-body raised is specific for SaARF4. It is possible that the data collected is a result of non-specific binding. This information should be included in the manuscript somewhere and further experiments should be conducted to determine the specificity of the anti-body.
I found the presentation of several figures and their legends to be confusing. It is not clear why Figure 1A and B are in the primary figure. They are discussed in the introduction and not in the results as new data. This data also does not directly connect to the rest of the experiments presented. While I appreciate the authors efforts to justify the use of the model plant, but this information confuses from the main point of the paper which is the relationship between auxin and ethylene as it relates to lateral and adventitious root formation. No further experiments assay Cd accumulation.
Figure 2G is mentioned in the text but not in the figure legend. In addition there is no information on the number of biological replicates or statistical tests used.
Figure 2B is difficult to read. The small model is not explained and the pictures appear to be very similar to one another. DR5 staining appears to be similar in these pictures. This may be a function of their quality in this version of the manuscript, but it is difficult to evaluate.
Figure 4A is poorly presented. It should match the presentation used in Figure 3C and D. The overlapping bars make it look like there is no change in ACO4 expression. In addition, there are no error bars presented in this data.
Figure 4D and E would be more compelling if they were presented together. It is difficult to see the effect of the AuxRE mutation in its current presentation.
Author Response
Dear reviewer:
Thank you for your comments. Our responses have been listed in the attachment. Please see the attachment.

Reviewer 2 Report
Review of the manuscript ‘Auxin-induced SaARF4 downregulates SaACO4 to inhibit lateral root formation of Sedum alfredii Hance’ by Xu et al.
Sedum alfredii is a Cadmium over accumulator and Xu et al have investigated the role of one of its ARF genes SaARF4 in regulating root development.
Over expression of SaARF4 results in fewer lateral roots a phenotype that is also caused by blocking ethylene synthesis pharmacologically using ethylene biosynthesis inhibitor PZA. They further show that expression of several PIN genes is significantly down in over expression lines and the same can also be mimicked by treatment of WT plants with PZA. PZA inhibits activity of ACOs to block ethylene production and authors also show that expression of ACO4 is significantly downregulated in SaARF4 over expression lines.
Based on these results and SaARF4 reporter studies as well as some experiments using DR5 reporter and dual luciferase and also ChIP assays authors conclude that ‘SaARF4 induced by auxin can inhibit ethylene biosynthesis by repressing SaACO4 expression, and this process may affect auxin transport to influence LRs development.’
I find this an interesting study and the research is of general interest to plant biologists. However, I find that there are some flaws in the approach, experimental design and interpretation of results. One of my main concerns is on functional significance. All the results are based on over expression of ARF4 which is not ideal as we can never be certain of the unintended ectopic over expression of a key transcriptional regulator. But at the same time, it is interesting that the over expression phenotype can be phenocopied by treatment with PZA. It is also very intriguing that despite several fold increase in ACO5 and ACO4.2 genes, (Fig 4A) there is a significant reduction ethylene production in OE lines which does not proportionate to the level of ACO4 downregulation. It is possible that ACO4 is the main ACO enzyme and hence its down regulation can have a serious impact on ethylene production, but if this is the case it is not clear and needs to be discussed. Not sure how many OE lines were used for study and if only one line has been used then this is not ideal and authors should report results of a few more lines. Same for ACS4::GUS lines and DR5 lines. DR5 lines have been independently introduced in WT and OE backgrounds. This is not ideal as the results could be misinterpreted due to position effect. Best will be to cross the WT DR5 line (that shows a 3:1 segregation ie one insertion) to the OE line and then compare the results once homozygous. In addition, there are several flaws in writing that need to be improved. I suggest using the term SAARF4 OE than just OE.Line 93 I don’t like the use of the term ‘unfortunately’. A result is a result. Line 113 ‘SaARF4 was expressed in vascular tissue and upregulated by auxin’Change was to is Line 114 ‘LAA sites are the places where LRs generate’.Not all local auxin accumulation or (LAA as the authors abbreviate) result in LR formation. Line 119 ‘Genes’ or gene Line 121 no need of ‘Compared with those of WT’ Line121 ‘Staining’ not stained Line 136 No need of ‘more’ Figure 2 No need for panel A and the legends need to be revised (remove ‘compared with WT’). Line 159 Rephrase the sentence ‘PIN genes-------root tips’ Not accurate to say this. Its not PIN genes but PIN proteins. Also they regulate auxin transport in other parts too. Fig 3 No need for ‘-100uM PZA’. -PZA is sufficient. Fig 3F Move the description of the bars from side to the top of figure 3F. Fig 3 Line 184. It is inaccurate to say ‘Promoter of DR5 were introduced’ so please rephrase.
Author Response
Dear reviewer:
Thank you for your comments. Our responses have been listed in attachment. Please see the attachment. Thanks!

Round 2
Reviewer 1 Report
The authors have made great progress in improving their manuscript. I have one major issue and a few minor issues.
Major Issue:
I appreciate the authors expanding upon their methodology and conducting controls for their SaARF4 anti-body. However, the authors neglected to include a western blot using one of the many miR390-OE lines they have. If their SaARF4 anti-body is specific, the miR390-OE lines should show greatly reduced or perhaps even absent SaARF4 protein. In addition, these westerns lack any loading control nor are they presented in the supplemental material, only in the response to reviewers. This may seem like a nitpick, but the ARFs share a consensus binding site. If their anti-body binds any other ARF it may result in a misinterpretation of the causal link between SaARF4 binding and ACO expression. The further experiments conducted in Figure 4 rely on direct binding of SaARF4 to the upstream region of ACO. If this result is in question, then it undercuts the wonderful work presented in the figure.
Minor Issues:
- Make mention of the role of Cd stress on LR number in the section related to Figure 4 so the reader can better interpret the results
- An explanation for the use of leaf tissue in the ChIP experiment as opposed to roots would be helpful. This experimental choice does not invalidate the conclusions drawn by the authors but it may invite more questions.
- The paper still needs modest amounts of English language revision but there are no large structural issues.
Author Response
Dear reviewer:
Thank you for your comments. We believe that your comments play important roles in improving our manuscript. Please see the attachment containing our response to your comments. Thanks again!

Reviewer 2 Report
Authors have addressed all my concerns in a reasonable manner and so I have no major concerns. I would still have preferred transgenic lines be created into WT background and then crossed into the OE lines but authors have used multiple lines with similar phenotypes.
Author Response
Dear reviewer:
Thank you for your comments. we believe that your comments play important roles in improving our manuscript. Please see the attachment containing the response to your comments. Thanks again!

Round 3
Reviewer 1 Report
The reviewers have fully addressed my concerns. I congratulate the authors on their manuscript.